# A group B *Streptococcus* indexed transposon mutant library to accelerate genetic research on an important perinatal pathogen

Venkata H. Bhavana,[1] Gideon H. Hillebrand,[2] Kathyayini P. Gopalakrishna,[1] Rebekah A. Rapp,[1,3] Adam J. Ratner,[4,5] Hervé Tettelin,[6] Thomas A. Hooven[1,2,7,8]

**ABSTRACT** Group B *Streptococcus* (GBS) is a major contributor to sepsis, meningitis, and pneumonia in newborns. Indexed bacterial mutant libraries accelerate pathogenesis research by allowing rapid screening of genes that contribute to disease. In this study, we created and characterized a large-scale GBS indexed library of *Himar1* mini-transposon mutant strains grown as monocultures from a mixed transposon insertion library that we had previously used for transposon-genome junction sequencing. We used a high-throughput workflow to identify transposon insertion sites in chromosomal DNA purified from individual mutants. Following quality control steps and the removal of isogenic duplicate strains, we isolated 1,919 monocultures of unique transposon insertion mutants with even dispersion across the GBS genome. Our final library, stored in barcoded, traceable glycerol stocks, contains interruptions of 878 genes and 253 intergenic regions. We also validated select library mutants with confirmatory PCR and, when possible, specific phenotypic testing. While the library contains only sparse interruptions of essential or near-essential genes, the genes represented in the set span a wide range of predicted functional categories, including roles in metabolism, structure, and virulence. In conclusion, we developed and applied a high-throughput molecular analysis and bioinformatic pipeline to generate a GBS indexed library of unprecedented scale that we believe will be a useful genetic tool for fellow GBS researchers.

**IMPORTANCE** Group B *Streptococcus* (GBS) is a significant global cause of serious infections, most of which affect pregnant women, newborns, and infants. Studying GBS genetic mutant strains is a valuable approach for learning more about how these infections are caused and is a key step toward developing more effective preventative and treatment strategies. In this resource report, we describe a newly created library of defined GBS genetic mutants, containing over 1,900 genetic variants, each with a unique disruption to its chromosome. An indexed library of this scale is unprecedented in the GBS field; it includes strains with mutations in hundreds of genes whose potential functions in human disease remain unknown. We have made this resource freely available to the broader research community through deposition in a publicly funded bacterial maintenance and distribution repository.

**KEYWORDS** *Streptococcus agalactiae*, transposon, mutagenesis, indexed library

*S*treptococcus agalactiae (group B *Streptococcus*; GBS) is a Gram-positive pathobiont found colonizing 20%–30% of healthy adults, where it is usually a minor subcomponent of the gastrointestinal and genitourinary commensal microbiota (1–3). Infections occur mostly in neonates and infants, among whom GBS is a significant contributor to bacteremia with resultant sepsis, pneumonia, meningitis, and joint and soft tissue infections (4–10). GBS vaginal colonization during pregnancy can develop into ascending chorioamnionitis (infection of the placenta, fetal membranes, amniotic fluid, and fetus),

Address correspondence to Thomas A. Hooven, thomas.hooven@chp.edu.

The authors declare no conflict of interest.

See the funding table on p. 15.

which causes maternal systemic infections, miscarriage, and preterm labor (3, 11–14). While rarer, GBS infection can also occur in otherwise healthy older children and adults, among whom case incidence has increased in the past decade, particularly in certain geographic regions (10, 15).

The World Health Organization has identified GBS as an important topic of research focus, with the goal of developing a widely available vaccine to prevent the infectious complications listed above (16). Multiple laboratory models of GBS colonization and pathogenesis exist (13, 17–21), which offer avenues to explore host-pathogen interactions that may contribute to disease. Genetic manipulation of GBS remains challenging, however. Strategies for targeted gene or promoter disruption in GBS require recombinant molecular biology techniques, including cloning, the generation of electrocompetent bacteria for plasmid transformation, and multiple rounds of passaging and screening to isolate the desired mutant (22, 23).

Indexed bacterial mutant libraries, consisting of stocked monocultures in which specific genetic alterations have been defined, can accelerate research, permitting rapid testing of specific gene contributions to disease (24–38). They can also be used to create curated sub-libraries for multiplex testing of gene sets that share common features, such as cellular localization or participation in particular metabolic pathways (29, 39). Currently, there is no publicly available GBS indexed library, which slows the pace at which GBS gene functions can be characterized and their contributions to disease determined.

In this report, we describe the creation and characterization of an indexed library of 1,919 monocultures of transposon insertion mutants in the GBS serotype Ia strain A909. This indexed library was derived from a mixed transposon insertion library, which we have previously used for transposon-genome junction sequencing (Tn-seq) to determine the essential genome of GBS and to identify genes that contribute to fitness in stringent host environments such as human blood and amniotic fluid (40–42).

Our A909 indexed library contains transposon insertions in 878 genes and 253 intergenic regions. We detail the process we used to identify transposon insertion sites in the library strains, outline genotypic and phenotypic validations we performed on a subset of the library, and offer recommendations for experimental use of the collection, which is made available via the NIH-funded BEI Resources (managed by the American Type Culture Collection) to other research teams in hopes that it will help advance efforts to eradicate GBS disease.

## RESULTS

### Development of a high-throughput method for indexing a GBS transposon library

We previously described using a plasmid-delivered *Himar1* mini-transposon to generate a mutant library in the serotype Ia GBS strain A909 (40). In that publication, we detailed the creation of pCAM48 (the plasmid we used to introduce the transposon system), the mutagenesis and quality control steps we took to ensure successful transposition and subsequent curing of pCAM48, and the next-generation sequencing analyses we performed to characterize the resultant libraries of mixed mutants.

The preparatory work generated three separate mutant libraries—subsequently combined into a "master" library—among which we detected 167,684 unique insertions of the transposon into the chromosome. The 1,455 base pair (bp) *Himar1* transposon that we used encodes an erythromycin resistance cassette and integrates randomly at TA dinucleotide sites in approximately one cell out of $10^5$. Therefore, the odds of any GBS cell bearing more than one transposon insertion are low. Similarly, while transposon insertion requires a transposase, which was co-delivered on pCAM48, spontaneous excision by homologous recombination along its 29-bp terminal inverted repeat segments is exceedingly unlikely. Growth in erythromycin-containing selective media is therefore not required for the propagation of mutants from our library.

Because of the transposon's size, insertion into a chromosomal coding sequence generally results in severe alteration of the transcript and functional loss of the gene. This is especially true for insertions that occur around the middle of a gene, as opposed to near its termini. Transposon insertion into an operon is also likely to disrupt the proper expression of polycistronic genes downstream of the insertion site. Intergenic insertions could also have a significant deleterious effect on the proper regulation of adjacent coding regions.

We have used the mixed A909 library described above in several Tn-seq studies. Its initial application, to determine the essential genome of A909 under laboratory outgrowth conditions, yielded a set of essential and near-essential genes that closely matched orthologous gene sets from Tn-seq studies in *Streptococcus pyogenes* (40, 43). Subsequent use of the library with Tn-seq to examine GBS genes with a significant contribution to fitness in human blood (41) and amniotic fluid (42) has yielded results that were validated through targeted allelic exchange mutagenesis approaches.

We developed a high-throughput strategy for identifying transposon insertion sites in chromosomal DNA purified from monocultures of mutant GBS grown from single colonies (Fig. 1). Culture outgrowth, DNA purification, and subsequent molecular biology steps were performed in 96-well sample plates. Prior to DNA extraction in a robotic liquid-handling and magnetic bead-based purification instrument, each 96-well monoculture outgrowth plate was replicated into a second 96-well plate, which—after overnight growth—was frozen following the addition of glycerol cryoprotectant.

Purified DNA was digested with the type IIS restriction enzyme MmeI, which generates an asymmetric, 2-nt overhang an average of 20 bp away from its recognition site. Because our *Himar1* transposon has MmeI sites integrated into its terminal inverted repeat sequences, digestion leads to 16-bp, GBS-specific, transposon-adjacent DNA sequences that can be used to identify the precise chromosomal insertion site in the source mutant.

We ligated double-stranded DNA adapters barcoded with 8 bp indicators of the well position from which the DNA originated. We then pooled the barcoded DNA and ran an initial PCR using primers matched to the transposon and the invariant portion of the ligated adapter. The transposon-specific primer was barcoded with an 8-bp code unique to the source plate. Next, all samples were pooled and subjected to a second round of PCR using outer primers that included overhanging Illumina flow cell binding sequences. The final 255-bp amplicon pool was purified from an agarose gel. Samples were then entered into a quality control and Illumina sequencing protocol described in Materials and Methods.

Demultiplexed sequences from our samples were decoded for plate and well position information from the barcodes. GBS-specific sequences identified based on adjacency to invariant transposon sequences were isolated and aligned to the A909 genome. We performed several quality-control screens on data from each well, including determination that the transposon insertion site was at a TA dinucleotide pair; the GBS-specific sequence between the transposon and the adapter was approximately the correct length [all sequences were 15–17 bp, reflecting previously recognized variation from MmeI digestion (44) and subsequent adapter ligation]; and that any alternative sequences from the well accounted for less than 10% of the reads. We then calculated a "confidence score" for every sample in our collection, which is the percent of filtered reads from that sample that map to the same A909 genome site. See Materials and Methods and Fig. S1 and Supplemental Data 1 for details about the DNA processing steps.

## Sequencing of 8,640 samples and selection of the indexed library strains

We performed three separate Illumina sequencing runs of sample collections that differed in size.

The first run comprised 11 sample plates (with 1,056 processed samples) and was intended to generate a preliminary data set to establish the feasibility of our pipeline

## Phase 1

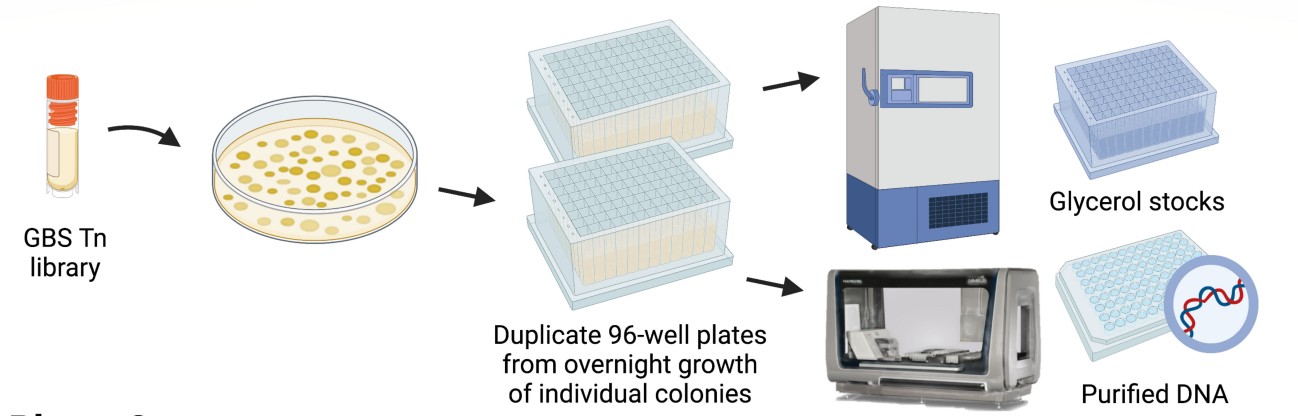

## Phase 2

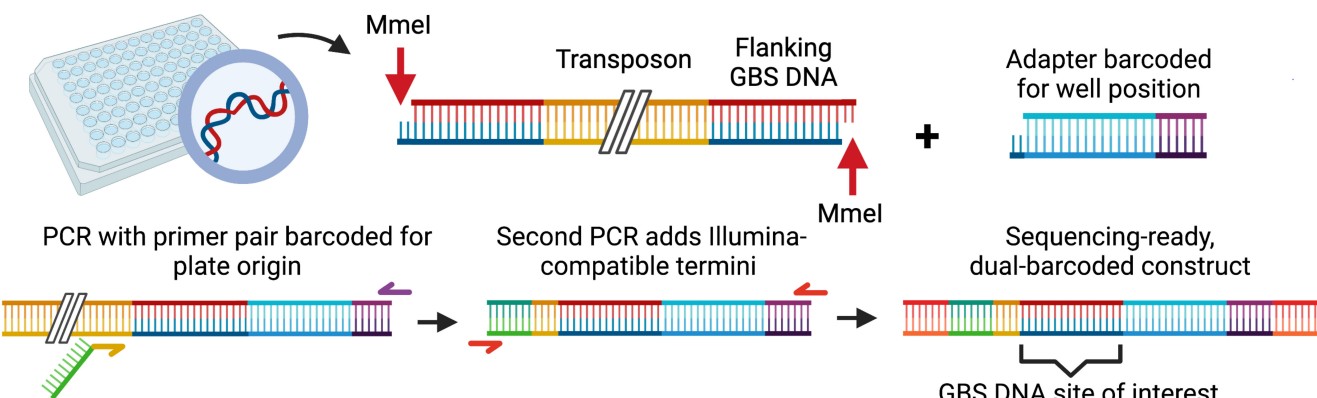

## Phase 3

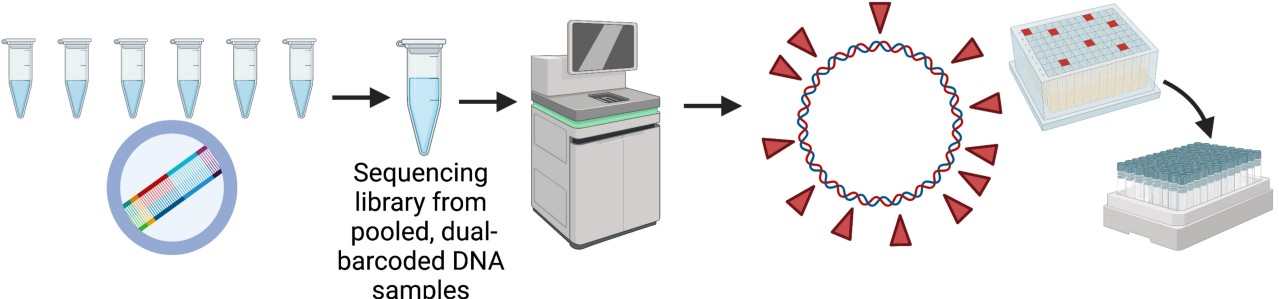

**FIG 1** Pipeline for group B *Streptococcus* strain A909 indexed library. In phase 1, an aliquot of the intermixed A909 transposon mutant library is used to inoculate a solid media plate, resulting in well-separated colonies, each with its own transposon insertion. Colonies are selected randomly and grown in duplicate, deep-well, 96-sample plates. One of the two plates is used for robotic DNA purification, while the duplicate is stored as a frozen stock source. In phase 2, the purified DNA from phase 1 is processed for sequencing and demultiplexing. MmeI digestion of the GBS transposon mutant DNA results in a genomic DNA sequence flanking the known *Himar1* transposon sequence. This allows the annealing of adapters barcoded for well position, followed by PCR that adds a plate-specific barcode and Illumina sequencing-compatible termini. In phase 3, pooled DNA preparations are deep sequenced and demultiplexed to identify the genomic insertion position for mutant strains in the collection. Barcode information is used to retrieve the correct stock plate wells, which are aliquoted into final library arrays using a traceable, robotic-assisted protocol.

and the interpretability of sequencing data. While this sequencing run did allow the effective identification of specific mutants, it also produced two unexpected findings that required interpretation. For some wells, we noted that the dominant transposon-adjacent sequence aligned not to the A909 chromosome but to the pCAM48 plasmid, often with a secondary transposon-adjacent sequence that aligned to the A909 genome. We believe that these findings were due to GBS samples that either had not cured the pCAM48 plasmid or in which the plasmid had integrated into the chromosome (possibly in addition to one or more transposon insertion events). We therefore excluded any such wells from the final indexed library.

We also noted that some well positions yielded low-level signals from the same A909 insertion across different plates and that these "echo" reads were the primary read from one well—of the same position—in one of the plates. The only explanation we could establish for this phenomenon was that trace amounts of DNA had accidentally contaminated the well-specific barcoded adapters (for instance, from inside a multi-channel pipette), leading to subsequent PCR amplification and detection in the highly sensitive sequencing instrument. We therefore bioinformatically removed these minor contamination reads from our analyses.

With a settled strategy for data screening and analysis using the first 11 plates (Set 1), we processed a batch of 59 sample plates (Set 2). Analysis of the resulting data demonstrated that the total number of unique insertions in our set continued to increase, but that the rate of increase was gradually slowing as discovering new insertions became less likely. We processed 20 additional plates (Set 3) and found that the rate of new insertion discovery had continued to level and was approaching an asymptote beyond which new insertions would become rare (Fig. S2).

From the set of 90 processed plates, we identified a total of 1,919 unique insertions detected in wells without errant primary reads from pCAM48, abnormal GBS DNA fragment length, or a confidence score below 0.7 (Supplemental Data 2). Many of the samples meeting these criteria were identified multiple times. When we had multiple samples with the same genotype, we retained the sample with the highest confidence score. In cases where the confidence score was equal (usually 1.0) across multiple identical samples, we retained the sample with the greatest number of sequence reads mapped to the assigned insertion. Figure 2 shows key metrics for the indexed library, including an illustration of relative proportions of coding sequence and intergenic transposon insertions in the library (Fig. 2A), sequence reads analyzed per mutant (Fig. 2B), and confidence indexes of library mutants (Fig. 2C).

Transposon insertions in the indexed library are evenly distributed across the genome (Fig. 3). Not surprisingly, interruptions of genes with high contributions to fitness in laboratory outgrowth were rare. Figure 2D shows the distribution of coding sequence insertions as a function of gene contributions to A909 fitness. The x-axis of the chart in Fig. 2D, labeled "Contribution to Fitness," reflects the $\log_2$ fold-change of observed-versus-expected insertions when the library was analyzed by Tn-seq (45). Lower values indicate higher gene contributions to fitness (i.e., more essential), while genes with high observed-versus-expected values have a lower contribution to fitness (i.e., less essential). Genes with very high observed-versus-expected Tn-seq values may negatively affect fitness during laboratory outgrowth, such that deleting them leads to improved mutant survival relative to wild-type competitors. Eleven genes (1%) in our indexed library were initially characterized as essential. We used targeted PCR to confirm these insertions (Fig. S3). We found that the transposon insertions within essential genes uniformly occurred very close to the 3′ terminus of the coding sequence. Among these 11 library members, the transposon insertion occurred between 93% and >99% along the open reading frame (median 99%). These 3′ terminal insertions likely resulted in a functional gene product, resolving the apparent paradox of a significant mutation in an expected essential gene. The remaining 867 interrupted genes were originally characterized as nonessential (40).

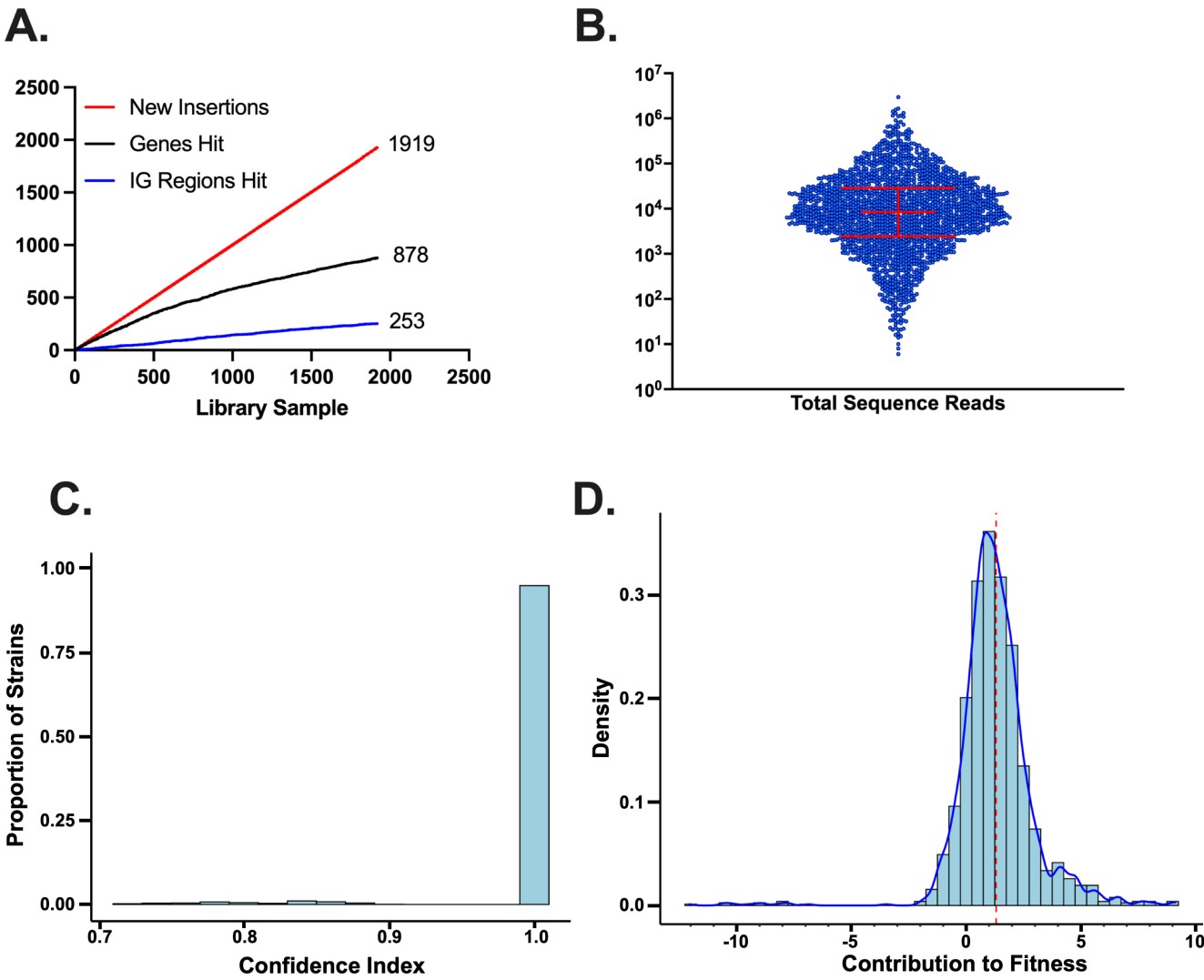

**FIG 2** Indexed library summary statistics. Number of gene (black line) and intergenic region (blue line) transposon insertions detected with each new well sequenced. Only the final library members are shown. Insertions at a different site within the same gene locus are added to the total new insertions plot (red line) only, demonstrating that there are multiple distinct insertions within many genomic loci (A). Number of sequencing reads mapping to the final insertion assignment for all 1,919 members in the library (B, red bars show median and interquartile range). Binned confidence indices for members of the library (C). Predicted contribution to the fitness of interrupted genes based on Tn-seq data. A value of zero indicates no predicted contribution to fitness under laboratory outgrowth conditions. Lower values indicate a greater contribution to fitness (more essential); higher values indicate less of a contribution (more dispensable, D); the red dotted line indicates the mean contribution to fitness value (1.31).

Once we had selected the set of mutants to comprise our final library, we retrieved the frozen glycerol stocks from the duplicated outgrowth plates saved at the beginning of our pipeline. We used a robotic liquid handling system running custom software to aliquot these stocks into four identical arrays of cryovial tubes in 96-sample racks. Each tube and rack are individually laser-etched with a one-dimensional (racks) or two-dimensional (tubes) barcode readable with many commercially available barcode scanners. The pipetting software we used for this project generates real-time sample tracking logs to ensure the long-term traceability of every sample in the collection back to its original source and sequencing data, all of which are publicly available.

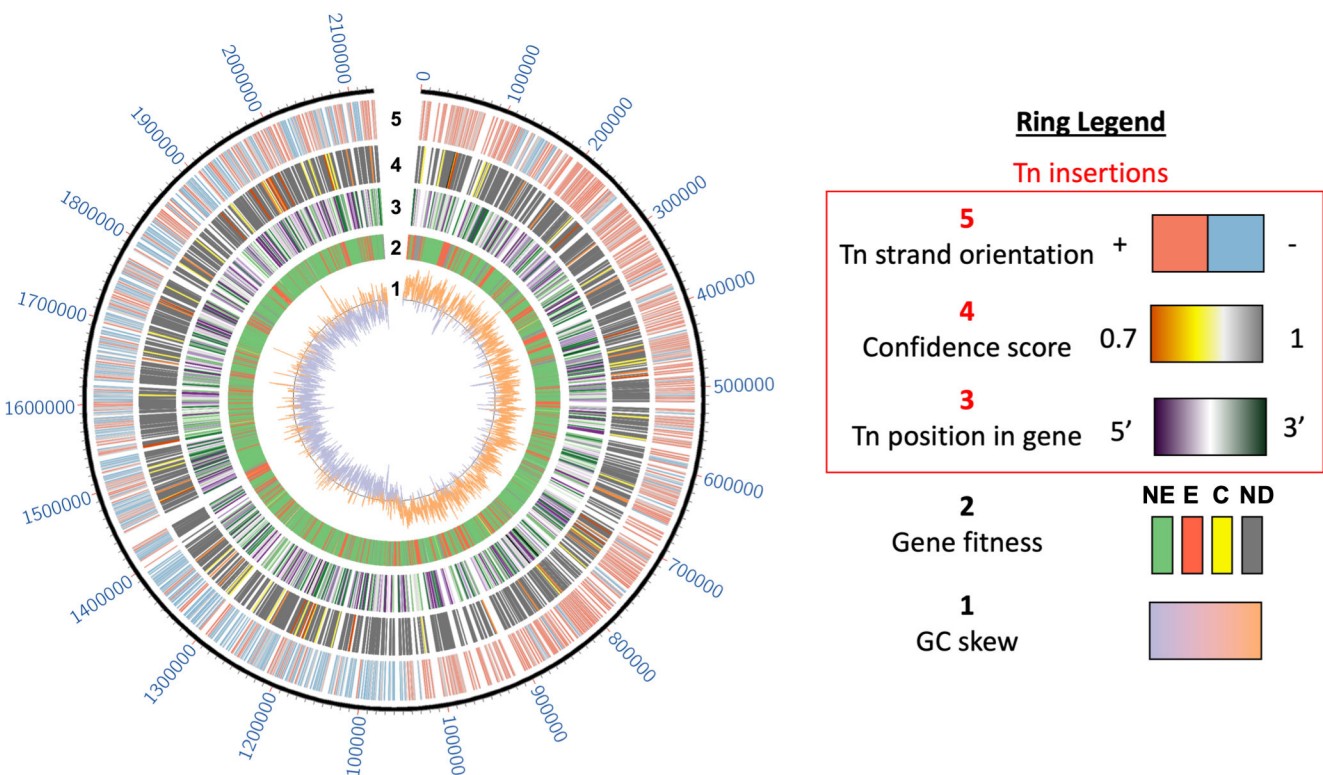

**FIG 3** Circos plot of indexed library insertions. Ring 1 shows a chromosomal GC skew. Ring 2 shows Tn-seq gene essentiality assignments based on prior quantification of the entire intermixed transposon library (E, essential; NE, nonessential; C, critical; ND, not defined); see reference (40). Rings 3–5 show the same 1,919 transposon insertions in the final indexed library with color-coded metrics as indicated in the legend.

## Genotypic and phenotypic confirmation of select indexed library strains

Several transposon insertion mutants in the library are predicted to have readily testable phenotypes. These include mutants with insertions in the *cyl* operon, whose activity is responsible for biosynthesis of the orange pigmented cytolytic toxin β-hemolysin/cytolysin (19, 46–48); the *cps* operon, which encodes enzymes that assemble and export the GBS polysaccharide capsule (49–52); and the *cfb* gene that encodes the Christie-Atkins-Munch-Peterson protein (CAMP factor), which causes enhanced β-hemolysis on sheep's blood agar plates when GBS is grown in proximity to *Staphylococcus aureus* that expresses toxin B (53, 54).

We studied these specific mutant strains, testing *cyl* and *cfb* activities by examining β-hemolysis on blood agar (growth was adjacent to *S. aureus* for the *cfb* assay) and *cps* activity using a commercially available GBS capsular serotype Ia latex agglutination kit. Comparisons were to wild-type A909 in all cases. Three insertion mutants in the *cyl* operon, three in the *cps* operon, and one in *cfb* showed the expected phenotypes, as shown in Fig. 4A through C. The *cyl* mutants were minimally or nonhemolytic; the *cps* mutants did not latex agglutinate; and the *cfb* mutant did not demonstrate hemolytic enhancement in proximity to *S. aureus*.

We also performed PCR verification of several transposon insertions, including insertions in the previously predicted 11 essential genes that were identified in the library (Fig. S3) and 12 other randomly selected genes whose disruption was not predicted to have distinct phenotypic effects (Fig. 4D).

## Screen for mutants with biofilm and growth effects

As a demonstration of how the indexed library can be used to rapidly screen curated panels of mutant strains for specific phenotypes, we tested a set of mutants with

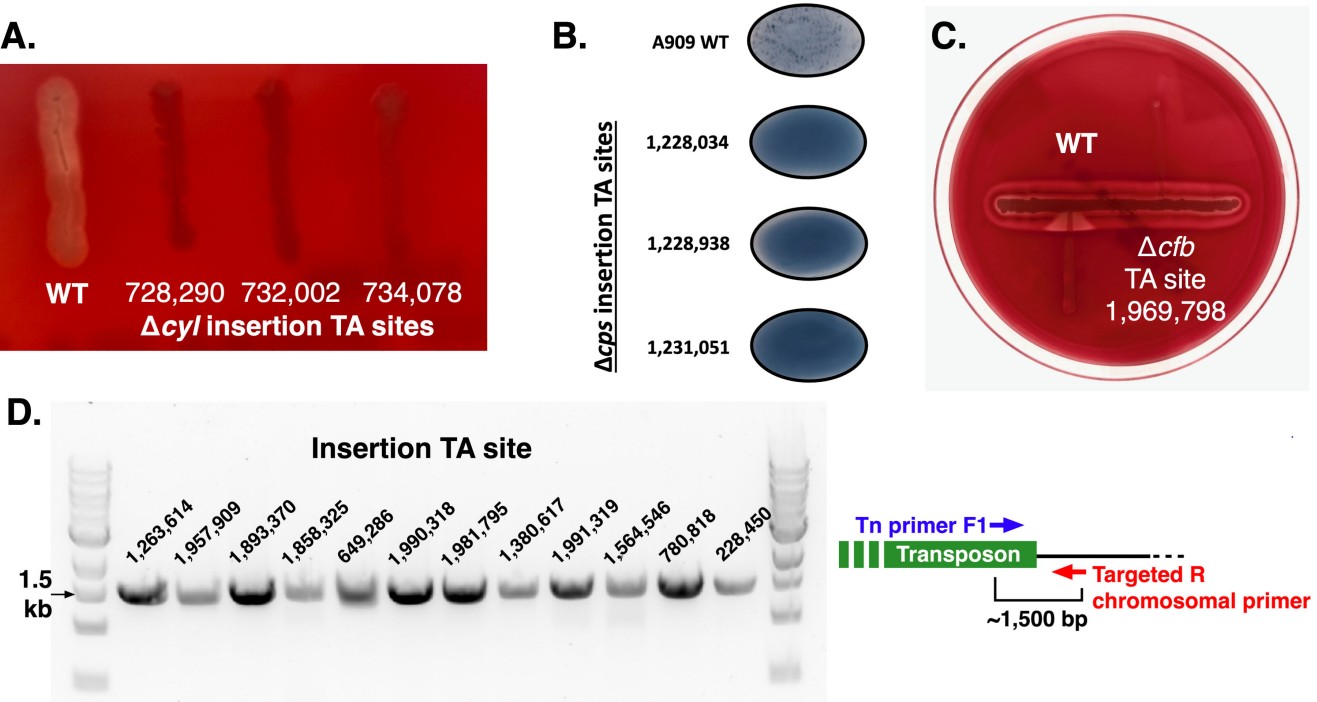

**FIG 4** Phenotypic and genotypic validation. Three mutants with transposon insertions in the A909 *cyl* operon, responsible for encoding enzymes in the biosynthetic pathway of the hemolytic pigment β-hemolysin/cytolysin, were plated and photographed, along with wild-type (WT) A909, on tryptic soy agar with 5% sheep's blood (A). Three mutants with transposon insertions in the *cps* operon, responsible for encoding enzymes in the biosynthetic pathway of the GBS polysaccharide capsule, were assayed with serotype Ia latex agglutination and photographed, along with wild-type A909; a positive result, indicating the presence of a normal capsule, is indicated by the formation of dark agglutination granules (B). One mutant with a transposon insertion in the *cfb* gene, responsible for a hyperhemolytic phenotype in the presence of *Staphylococcus aureus*, was assayed and photographed next to the control wild-type A909 (C). A positive result appears as an "arrowhead" of enhanced hemolysis, as seen on the wild-type streak. Genomic DNA from 12 transposon insertion mutant strains was screened by PCR to confirm expected insertions (D). The PCR design is indicated in the inset diagram. In all panels, the mutants assayed are indicated by the TA site of the transposon insertion.

transposon interruptions to signal peptide-encoding genes in the GBS core genome for effects on biofilm formation and growth rates. Well-characterized signal peptide motifs at the N-termini of proteins are used by bacteria as trafficking flags, marking them for externalization—either surface anchoring or secretion—by enzymes in the bacterial Sec trafficking pathway (55). We reasoned that these externalized proteins, when deleted, may lead to previously uncharacterized effects in an *in vitro* biofilm assay and in standard measures of growth kinetics.

To identify core genome genes with signal peptides, we screened 671 complete GBS genomes posted to the Joint Genome Institute's Integrated Microbial Genomes and Microbiomes database for all genes marked as containing signal peptides (56). The database allows cross-tabulation of signal peptide-bearing genes, allowing identification of genes shared among all 671 genomes with at least 10% homology in their DNA coding sequences. In this way, we identified 70 core genome genes with signal peptides. We assessed each of the 70 genes using SignalP software (57) to confirm the presence of predicted signal peptide sequences, which verified the presence of the predicted trafficking tags (Fig. S4).

Our indexed library contained 29 mutants with predicted deletions of core signal peptide genes. Assaying for biofilm formation in plastic, flat-bottomed multi-sample cell culture plates revealed three mutants with significant differences in biofilm formation. Wild-type A909 is a relatively poor biofilm-producing strain (58), and we were interested to see that the mutants we identified in our screen with biofilm alterations

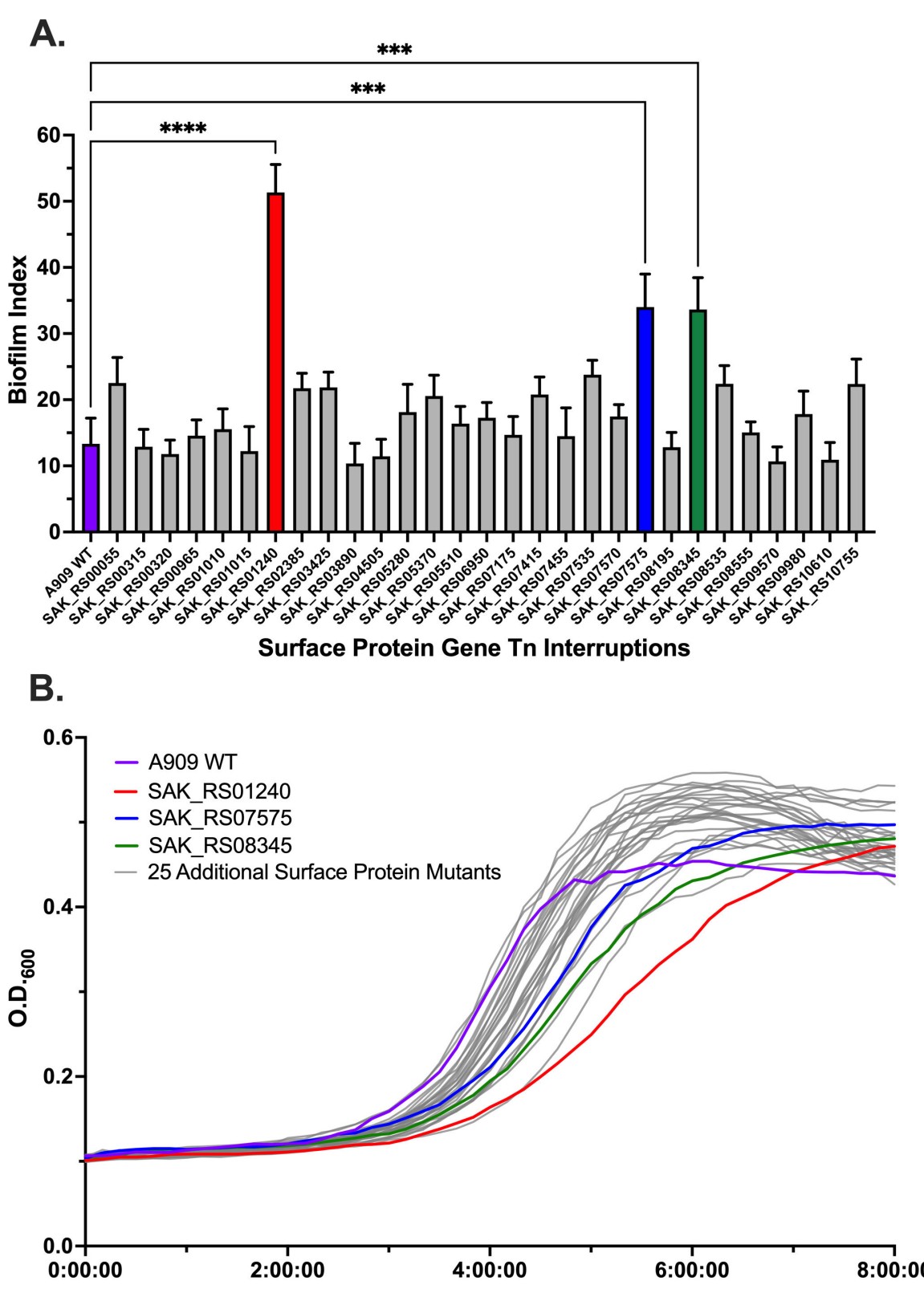

**FIG 5** Phenotypic testing of GBS core signal peptide-encoding gene mutants. Twenty-nine mutants with transposon insertions in core GBS signal peptide-encoding genes were tested in *in vitro* biofilm assays (A, bars show mean values with standard error of the mean; ****$P < 0.001$, ***$P < 0.005$, analysis of variance with Dunnett's correction) and 8-hour growth curves (B, plots show mean values across replicates). Each assay was conducted on three separate biological replicates.

produced significantly more substantial biofilms than the wild-type control (Fig. 5A). The three genes we identified are predicted to encode an amidase (SAK_RS07575), an ABC transporter (SAK_RS08345), and a hypothetical protein of uncertain function (SAK_RS01240). Assessing the growth kinetics of the panel of signal peptide gene mutants revealed growth delays in each of the three mutants with biofilm effects (Fig. 5B), suggesting that the genes interrupted in these three mutants may play significant external structural roles that manifest as efficient growth and dispersal during static planktonic division.

## DISCUSSION

While GBS remains a dominant contributor to serious perinatal disease, the challenges associated with generating targeted mutants create a bottleneck that slows new discoveries about its pathogenic mechanisms.

Indexed mutant libraries have accelerated research on other major human pathogenic bacterial species. One of the first—and most widely used—indexed bacterial mutant libraries is the Keio collection of in-frame mutations in *Escherichia coli* K-12 (38). An important example of an indexed mutant library in an encapsulated Gram-positive bacterium is the Nebraska Transposon Mutant Library in *S. aureus* USA300 (37), which has been instrumental in hundreds of studies of *S. aureus* biology and infection. Indexed libraries, generated with a variety of approaches, have also been employed in the study of *Bacillus subtilis* (31), *Enterococcus faecalis* (29), *Pseudomonas aeruginosa* (26, 30, 59), *Salmonella enterica* serovar Typhimurium (34), *Proteus mirabilis* (33), *Bacteroides thetaiotaomicron* (32), *Borrelia burgdorferi* (27), and *Acinetobacter* spp. (24, 36).

We used a PCR- and sequencing-based approach to individually characterize 1,919 different transposon insertion mutants in our GBS A909 indexed mutant library, which is derived from a large, intermixed transposon mutant library we initially developed for Tn-seq. The serotype Ia A909 strain has a 2,127,839-bp chromosome (NCBI GenBank accession no. GCA_000012705.1) with 2,173 predicted genes, which were sequenced in 2005 (60). First described by Lancefield, McCarty, and Everly in 1975 (61), A909 can be considered among the "canonical" GBS strains in the original Lancefield strain collection at Rockefeller University. Its long history of extensive genetic study and use in experimental infection models motivated us to use this strain as a background for our indexed library, reasoning that findings associated with genes in A909 would be broadly interpretable within the context of its long history of study. Following the selection of final indexed library strains, we validated select stock strains genotypically by PCR to amplify expected junctions between chromosome and transposon sequences and phenotypically when the gene interruption was expected to yield a well-characterized phenotype.

However, most strains in the library were not individually validated after multiplex determination of the transposon sequence. Users are therefore advised to undertake reasonable validation protocols in their own laboratories when performing experiments with strains in the library. We recommend genotypic confirmation by PCR. Instructions for our protocols for mutant confirmation by PCR are provided in Supplemental Data 3.

One obvious use of the library is to study individual genes or intergenic regions of interest. The library includes interruptions of 878 genes and 253 intergenic regions. Multiple transposon interruptions to many of these loci are present in the library, providing potential redundant replicates for experiments focused on a particular gene or promoter region.

Additionally, the same Tn-seq approaches that were developed and validated with the full A909 intermixed library can be applied to subsets of the indexed library described here. The advantage of performing multiplex fitness testing on curated subsets of mutants—rather than a large, intermixed library—is that as the number of statistical comparisons between strains decreases, one's ability to detect subtle differences between strains increases. Furthermore, by starting with physically separated mutants, it becomes possible to ensure that the starting inoculum of each strain in a

curated sub-library is equal. Starting with equal inoculums of each mutant simplifies analysis of changes over time, making selective bottlenecks that affect one strain more than another easier to detect (62).

While extensive, this indexed library of A909 mutants is not a comprehensive collection of nonessential gene knockouts. Not surprisingly, we isolated relatively few mutants expected (based on Tn-seq data) to adversely affect bacterial fitness during laboratory outgrowth (see Fig. 2D). While transposon insertions within genes that are critical for efficient GBS outgrowth are detectable by very deep sequencing in the intermixed Tn-seq library, these low-fitness mutants were presumably outcompeted by the higher-fitness mutants during the growth of the library (not only during this study but also in the rounds of outgrowth that were necessary for initial library development) (40). In accordance with this model, we found that as we indexed more colonies, the chances of finding new gene mutants decreased with time until we reached a point at which the time and expense of further characterization were unjustifiable (see Fig. S2).

Nevertheless, we anticipate that this collection of indexed GBS mutants—unprecedented in scale and made freely available to the microbiological research community—will spur an acceleration in GBS pathogenesis discovery and further progress toward scalable, targeted approaches for GBS infection prevention and treatment.

## MATERIALS AND METHODS

### Bacterial strains and growth conditions

A909 is a sequence-type 7 GBS strain that expresses capsular serotype Ia. It was originally isolated from an infected neonate and has been used in many laboratory studies of GBS pathogenesis (22, 51, 61, 63–69). GBS was grown in liquid cultures in tryptic soy (TS) broth (Fisher Scientific cat. # DF0370-17) under stationary conditions at 37°C in atmospheric conditions (i.e., no $CO_2$ supplementation) without antibiotic selection. Bacterial colonies for monocultures were grown on solid TS medium with 15 g/L agar (Fisher Scientific cat. # DF0369-17) at 37°C in atmospheric conditions. *S. aureus* strain RN6930 (70) for CAMP factor detection was grown under stationary atmospheric conditions in a frozen glycerol stock solution on solid and liquid TS media at 37°C.

### Transposon mutant library outgrowth for single colonies

Frozen glycerol stock samples of previously described A909 *Himar1* transposon insertion mutant libraries A2, A5, and A7 (40) were combined in equal volumes, aliquoted into microcentrifuge tubes, and then replaced at −80°C. Prior to outgrowth and plating, a single tube was removed from frozen storage and thawed on ice, and 20 µL of the library stock was directly instilled into 10 mL of prewarmed TS broth at 37°C. After 4 hours, these samples were diluted 1:10 to $10^{-6}$, and then 100 µL of each dilution was plated on solid TS media overnight (night 1). On the next morning, we selected the dilution plate with colonies that were as numerous as possible while remaining distinct and separated enough for individual colony selection. We used sterile micropipette tips to individually select random colonies for transfer into 2 mL of liquid TS in 96-deep well format. The seeded plate was sterilely covered and allowed to grow at 37°C overnight (night 2). On the next morning, we used a flame-sterilized plate replicator device (Boekel Scientific cat. # 140500) to seed 600 µL TS in a second 96-deep well plate with grown culture from the first plate. The second plate was allowed to grow overnight (night 3), while the first was used for bacterial genomic DNA extraction (see below). On the next morning, the replicated plate had 200 µL sterile glycerol added to each well, after which it was again sterilely covered and moved to −80°C.

### Genomic DNA extraction

Genomic DNA was extracted from the 96-deep well overnight growth samples (from night 2) on a Hamilton Nimbus Microlab 4-channel automated liquid handling

instrument with an integrated ThermoScientific Kingfisher Presto multiplex purification instrument fitted with a 96-head electromagnet. The Nimbus-Presto was programmed to perform automated DNA extraction in 96-well format using reagents from the MagMAX Viral/Pathogen Ultra Nucleic Acid kit (ThermoScientific cat. # A42356) with the following modifications.

Lysis buffer from the kit was replaced with mutanolysin (Sigma cat. # M9901-50KU) and lysozyme solution in a potassium phosphate buffer. First, 280 µL filter-sterilized mutanolysin stock (10 kU in 2 mL of 0.1 M potassium phosphate buffer at pH 6.2) and 140 mg lyophilized lysozyme from chicken egg white (Sigma cat. # L4919) were dissolved in 720 µL 0.05 M potassium phosphate buffer at pH 6.2 with 0.25 M added NaCl. Once dissolved, 1 mL of sterile glycerol was added along with sterile RNase (Sigma cat. # R6513-50MG) to a final concentration of 50 mg/mL.

The wash buffer was replaced with 30 mL of 30% polyethylene glycol-6000 with 1 mL of 0.1 M Tris-HCl pH 8.0, 300 µL 1% Tween-20, 300 µL 0.1 M EDTA at pH 8.0, and 3 g guanidium thiocyanate (to make 4 M) added. The kit elution buffer was replaced with sterile 10 mM Tris-HCl at pH 8.0.

## Generation of double-stranded, well-specific barcoded adapters

Steps for this portion of the method were adapted from reference (70) with modifications. Custom synthesized, complementary, single-stranded oligonucleotides were purchased from Integrated DNA Technologies (Coralville, IA, USA). Each oligonucleotide was barcoded with an 8-nt sequence specific to one well position on a 96-well plate. The oligonucleotides were asymmetric, with one strand ending in a degenerate NN dinucleotide overhang, while the other strand was modified with 5′ and 3′ phosphate groups. The adapter design is intended to prevent self-ligation while permitting pairing with the 2-nt MmeI-generated sticky end (which can consist of any two bases) 20 bp upstream of the MmeI recognition site (located within the transposon inverted repeat). For each well position, a double-stranded adapter stock was created by combining the two single-stranded components and annealing them using standard techniques (70).

## Library genomic DNA processing for sequencing and indexing

Twenty-two microliters of genomic DNA was digested with *MmeI* (New England Biolabs cat. # R0637L) combined with 1× Antarctic phosphatase (New England Biolabs cat. # M0289L) and ZnCl 1 mM in CutSmart Buffer. The reaction was allowed to proceed for 3 hours at 37°C and heat-inactivated at 80°C for 2 minutes and 65°C for 20 minutes.

Then, 13.5 µL of digested genomic DNA was combined with T4 ligase (New England Biolabs cat. # M0202L), ATP (New England Biolabs cat. # P0756L), and a double-stranded, well-barcoded adapter. This reaction was allowed to proceed overnight at room temperature.

Next, 5 µL of ligation reaction from each well was pooled across an entire plate. These pooled samples were purified with 0.8× magnetic AMPure XP beads (Beckman Coulter cat. # A63880) according to the manufacturer's recommendations. Purified pooled samples were then used for PCR reaction 1 (see Supplemental Data 1), which used reverse primers barcoded for plate origin. Q5 Hi-fidelity 2X Master Mix (New England Biolabs cat. # M0492L) was used for both PCR reactions 1 and 2. The thermocycler settings were as follows: initial denaturation of 30 seconds at 98°C, followed by 25 cycles of denaturation at 98°C for 2 minutes, annealing at recommended $T_m$ (61°C—PCR 1, 67°C—PCR2) for 30 seconds, extension at 72°C for 10 seconds, and a final extension of 2 minutes at 72°C. The 186-bp PCR amplicon was gel extracted and used as a template for PCR reaction 2, which added Illumina-compatible flow cell binding sequences to the amplicon termini. The 255-bp PCR product was gel-extracted, and its size and concentration were measured by TapeStation analysis. A final pooled sample was prepared with equimolar concentrations of PCR product from each included sample plate. The recombinant DNA steps are illustrated in Fig. S1.

## Illumina sequencing

Three Illumina library pools were assessed for quality using the Perkin Elmer Gx Touch (Waltham, MA, USA), revealing a peak centered at 266–267 bp (there was some noise in the signal, which we believe accounted for the difference between our expected and measured amplicon size values). Quantitative PCR was used to assess the number of fragments suitable for sequencing in the library pool and determine the loading concentration on the Illumina NovaSeq 6000 (San Diego, CA, USA). Samples were sequenced on a 150-bp paired-end run and demultiplexed using custom pipelines. Illumina R1 reads obtained using the Illumina F sequencing primer were screened by pattern matching using a Perl script with the following regular expression: ([NACGT]{8}) ([NACGT]+)ACAGGTTGGATGATAAGTCCCCGGTCTGACACATAGATGGCGTCGCTAGTATTAAAT GCAGTAGATCCGAAGATCAGCAGTTCAACC([NACGT]{8})TCATAGATCGGAAGAG; where the first instance of ([NACGT]{8}) is the well barcode, ([NACGT]+) is the sequence of the GBS insertion site, the following sequence is from the pCAM48 plasmid, the second instance of ([NACGT]{8}) is the plate barcode in reverse complemented orientation, and the remaining sequence is the end of the Illumina R sequencing primer in reverse complemented orientation. Using in-house Perl scripts, screened reads were then grouped by identical plate and well barcode pairs, and those carrying the expected barcodes for the plates used in each set were retained (i.e., Sets 1, 2, or 3; plate barcodes were reused between the sets but not within any given set). The total number of reads obtained for each individual well and the number of each group of identical reads from the well were determined; then reads of 10 bp or longer representing 10% or more of the total number of reads obtained for the well were enumerated and ranked based on their abundance, with rank 1 being the most abundant read. Ranked reads that were present in the same well of multiple plates of the library ("echo" reads, see Results) were eliminated (and their number deducted from the well total) unless they were of rank 1. Finally, the remaining reads were mapped to the A909 genome (as well as the pCAM48 plasmid sequence as a control) using Bowtie 1.2.2, and their location with respect to GBS genes and intergenic regions was characterized using Samtools 1.11 and in-house Perl scripts. See Results and Supplemental Data 2-3 for additional quality control verifications and calculations performed.

For public data release, we generated three fastq files, one for each library or set (Set 1: 22,665,792 reads; Set 2: 52,616,824 reads; and Set 3: 80,148,220 reads), containing the raw Illumina versions of the reads selected using these procedures (whether or not they mapped to the A909 genome). See Data Availability.

## Thawing and aliquoting of library strains

Frozen glycerol stock plates from sequence-characterized mutant colonies were thawed at room temperature. Once a source stock plate was melted, a custom script was used to import CSV-encoded "pick lists"— indicating each well from the source stock plate selected for inclusion in the final library—into Hamilton Nimbus Microlab 4-channel robotic liquid handling software (version 4.6.0.7995). Next, the script drives the instrument to mix, aspirate, and aliquot 175 µL of thawed glycerol stock into four arrays of individually barcoded ThermoScientific Matrix cryovials arranged in 96-sample plates. The script also employs sample tracking capabilities to produce trace files for every sample in the library so that the samples in each barcoded tube can be definitively linked to a primary culture and sequencing result.

## PCR verification of select strains

We performed PCR on genomic DNA from select mutants in the library to confirm expected transposon insertion sites. We selected a panel of 11 mutants affecting genes previously characterized as essential and 12 mutants affecting nonessential genes. We used Tn Primer F1 and transposon-adjacent genomic DNA primers matched with Tn Primer F1 for predicted DNA binding properties (Supplemental Data 1 and 3). Q5

Hi-fidelity 2X Master Mix (New England Biolabs cat. # M0492L) was used for PCR. The thermocycler settings were as follows: initial denaturation of 30 seconds at 98°C, followed by 30 cycles of denaturation at 98°C for 2 minutes, annealing at the recommended $T_m$ for 30 seconds, extension at 72°C for 10 seconds, and a final extension of 2 minutes at 72°C.

## Visualization of hemolytic activity

Wild-type and mutant A909 strains were grown on solid medium from frozen stocks. Individual colonies were streaked onto TS agar plates with 5% sheep's blood (Thermo-Scientific cat. # R01200) using a sterile applicator. The streaks were grown overnight at 37°C and photographed the next morning.

## CAMP factor activity assay

The library mutant strain with an expected transposon insertion at nucleotide position 1,969,798, interrupting the *cfb* gene, was grown alongside wild-type A909 on a blood agar plate in proximity to *S. aureus* strain RN6930 (70) overnight. On the next morning, regions of proximity between GBS and *S. aureus* were evaluated for a visible increase in β-hemolysis (54).

## Latex agglutination for type Ia capsule detection

The ImmuLex group B *Streptococcus* type Ia kit (Cedarlane cat. # 54982) was used with an individual colony of wild-type or mutant A909, grown overnight, and suspended and vortexed in sterile phosphate buffered saline (PBS). The assay was performed according to the manufacturer's instructions. The agglutination reactions were photographed 1 minute after combination with resuspended A909.

## Identification of core genome signal peptide-encoding genes

*Streptococcus agalactiae* genomes posted to the Joint Genome Institute's Integrated Microbial Genomes and Microbiomes database ($n = 671$) were screened for genes with predicted signal peptides and presence with at least 10% identity among all the genomes. The FASTA sequences of the resulting 70 genes were then individually tested with SignalP. Amino acid summary probabilities for being part of a signal peptide, signal peptide cleavage site, or post-cleavage protein coding sequence are plotted in Fig. S4, which indicates that all the genes in the set do encode signal peptides.

## Biofilm assay

Wild-type A909 and mutant strain cultures were grown overnight in TS broth and sub-cultured at a 1:100 dilution into fresh media with 5% glucose supplementation. Diluted cultures were aliquoted into 24-well plastic flat-bottom plates, which were grown stationary at 37°C for 24 hours. After growth, the plates were gently rocked for 1 minute to resuspend nonadherent bacteria. Media with planktonic bacteria were gently aspirated and transferred to clean wells, and $OD_{600}$ was recorded. The remaining biofilms on the overnight plate wells were stained with 0.1% crystal violet for 7 minutes and gently rocked. The wells were then washed twice with PBS and allowed to dry. After the addition of an 80% ethanol/20% acetone solution, the $OD_{560}$ of the solubilized biofilms was recorded. The biofilm index was calculated by the ratio of $OD_{560}$ to the $OD_{600}$ measurement of the planktonic overnight cultures.

## Growth curves

Overnight GBS cultures were diluted 1:1,000 in fresh media and seeded into 96-well, clear plastic plates. $OD_{600}$ measurements were obtained every 10 minutes at 37°C, with 10 seconds of rotary shaking before each measurement, on an automatic plate reader.

## ACKNOWLEDGMENTS

We are grateful to William MacDonald, Rania Elbakri, and their staff in the University of Pittsburgh Health Sequencing Core for their assistance with DNA quality control measurements. Christian Hafele at Hamilton Company provided crucial help with the development and troubleshooting of custom robotic liquid handling scripts. We also wish to thank members of the Maryland Genomics Core within the University of Maryland School of Medicine for library QC and sequence data generation.

Illustrations in this paper were created with BioRender, Geneious, and Graphic software packages.

Financial support was provided by NIH/NIAID R21AI147511 to T.A.H., A.J.R., and H.T. The funders had no role in study design, data collection and analysis, the decision to publish, or the preparation of the manuscript.

## AUTHOR AFFILIATIONS

[1]Department of Pediatrics, University of Pittsburgh School of Medicine, Pittsburgh, Pennsylvania, USA

[2]University of Pittsburgh, Graduate Program in Microbiology and Immunology, Pittsburgh, Pennsylvania, USA

[3]The Ellis School, Pittsburgh, Pennsylvania, USA

[4]Department of Pediatrics, New York University, New York, New York, USA

[5]Department of Microbiology, New York University, New York, New York, USA

[6]Department of Microbiology and Immunology, Institute for Genome Sciences, University of Maryland School of Medicine, Baltimore, Maryland, USA

[7]Richard King Mellon Institute for Pediatric Research, University of Pittsburgh Medical School, Pittsburgh, Pennsylvania, USA

[8]UPMC Children's Hospital of Pittsburgh, Pittsburgh, Pennsylvania, USA

## AUTHOR ORCIDs

Thomas A. Hooven  http://orcid.org/0000-0003-1959-186X

## FUNDING

| Funder | Grant(s) | Author(s) |
| --- | --- | --- |
| HHS | NIH | National Institute of Allergy and Infectious Diseases (NIAID) | R21AI147511 | Thomas A. Hooven |
| | | Adam J. Ratner |
| | | Hervé Tettelin |

## AUTHOR CONTRIBUTIONS

Venkata H. Bhavana, Investigation, Methodology, Project administration, Validation, Visualization, Writing – original draft, Writing – review and editing, Formal analysis | Gideon H. Hillebrand, Investigation, Visualization, Writing – review and editing | Kathyayini P. Gopalakrishna, Investigation, Visualization, Writing – review and editing | Rebekah A. Rapp, Investigation, Writing – review and editing | Adam J. Ratner, Conceptualization, Data curation, Formal analysis, Funding acquisition, Investigation, Methodology, Resources, Writing – review and editing | Hervé Tettelin, Conceptualization, Data curation, Formal analysis, Funding acquisition, Investigation, Methodology, Software, Writing – review and editing | Thomas A. Hooven, Conceptualization, Data curation, Formal analysis, Funding acquisition, Investigation, Methodology, Project administration, Resources, Software, Supervision, Validation, Visualization, Writing – original draft, Writing – review and editing

## DATA AVAILABILITY

One complete mutant library for public access is stored at BEI Resources (beiresources.org). The accession number for the complete library is NR-59477. BEI accession numbers for individual 96-well plates and individual mutants within the library can be found in Supplemental Data 2. At the time of publication, BEI anticipates being ready to distribute the GBS library in April 2024. Sequence reads used to specify transposon insertion sites for strains within the library are posted to the NCBI Sequence Read Archive (SRA) under BioProject PRJNA1000625.

## ADDITIONAL FILES

The following material is available online.

### Supplemental Material

**Suppl. Data 1 (Spectrum02046-23-s0001.xlsx).** Primers and adapters.
**Suppl. Data 2 (Spectrum02046-23-s0002.xlsx).** Indexed library details.
**Suppl. Data 3 (Spectrum02046-23-s0003.docx).** Recommendations for validation.
**Suppl. Figure 1 (Spectrum02046-23-s0004.tif).** Amplicon construction.
**Suppl. Figure 2 (Spectrum02046-23-s0005.tif).** New gene discovery rate.
**Suppl. Figure 3 (Spectrum02046-23-s0006.tif).** Essential gene testing.
**Suppl. Figure 4 (Spectrum02046-23-s0007.tif).** Signal peptide analysis.
**Supplemental Data Captions (Spectrum02046-23-s0008.docx).** Aggregated captions for all supplemental figures and data.

### Open Peer Review

**PEER REVIEW HISTORY (review-history.pdf).** An accounting of the reviewer comments and feedback.

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
