## [Reviewer comments · Microbiology Spectrum]

Microbiology Spectrum

A group B Streptococcus indexed transposon mutant library to accelerate genetic research on an important perinatal pathogen

Venkata Hemanjani Bhavana, Gideon Hillebrand, Kathyayini Parlakoti Gopalakrishna, Rebekah Rapp, Adam Ratner, Herve Tettelin, and Thomas Hooven

Corresponding Author(s): Thomas Hooven, University of Pittsburgh

Review Timeline:

Submission Date:	May 18, 2023
Editorial Decision:	June 12, 2023
Revision Received:	September 8, 2023
Accepted:	September 29, 2023

Editor: Shannon Manning

Reviewer(s): Disclosure of reviewer identity is with reference to reviewer comments included in decision letter(s). The following individuals involved in review of your submission have agreed to reveal their identity: Michelle L. Korir (Reviewer #1)

Transaction Report:

DOI: <https://doi.org/10.1128/spectrum.02046-23>

June 12, 2023

Dr. Thomas A Hooven
University of Pittsburgh School of Medicine
Department of Pediatrics
Pittsburgh, PA 15224

Re: Spectrum02046-23 (A group B Streptococcus indexed transposon mutant library to accelerate genetic research on an important perinatal pathogen)

Dear Dr. Thomas A Hooven:

Thank you for submitting your manuscript to Microbiology Spectrum. Your paper has been reviewed by 2 experts in your field and both agree that the data are interesting, and the tool is important for conducting additional research on GBS. Nonetheless, the reviewers made several suggestions for modifying and improving your existing manuscript. Both noted the need for providing additional justification for the strain used as well as the importance of examining mutants that have not already been evaluated. Reviewer 1 mentioned that more details are needed throughout, while Reviewer 2 noted limitations associated with covering only ~50% of the genome. Keeping these and the remaining suggestions in mind, I would like to invite you to submit a revision for consideration.

Link Not Available

Sincerely,

Shannon Manning

Journals Department
Reviewer comments:

Reviewer #1 (Comments for the Author):

General Comments

The authors used a previously developed transposon mutant library that had been used in prior TN-seq experiments to generate an indexed transposon mutant library for use in future genetic studies in GBS. This paper presents an extremely useful tool to be

used by the research team and is open to the broader scientific community. As a proof of concept, the authors validated phenotypes of a select set of mutants with Tn insertions in previously studied genes. Although this was meant purely as a proof of concept, it would have been more impactful if mutants in genes that had not been thoroughly studied previously had also been examined to further show the usefulness of this tool and assign functions to genes. Additionally, certain portions of the manuscript need further clarification or additional details. More specifically:

Lines 42-43: in the sentence "while the library contains....." What exactly are you referring to here? Are you saying very few have growth defects? If so, this would be expected. Please clarify

In the introduction of bacterial mutant libraries (lines 74-78), or in another section of the introduction, please add information regarding transposon mutant libraries in GBS and the lack of an indexed library and its usefulness to fully explain to the reader the need for this paper.

Please provide more information on the background genetics of A909 such as: Why did you choose this particular strain? How large is the genome? How many genes, etc to determine which percentage of the genome and number of genes were hit with TNs in this library.

Line 160 mentions previously recognized variation from Mmel digestion, but no reference is provided.

Several areas of the paper are divided into rather short paragraphs, making the flow choppy and occasionally causing confusion. For example, in the paragraph for lines 173-176, you mention two unexpected findings, but then the paragraph ends leading the reader to first think you aren't going to mention them and then guess that what is discussed in the next paragraph are the unexpected findings.

Line 194 - I understand what you are trying to say by "slowing rate of new insertion discovery" but this phrasing is odd and could be confusing for someone less familiar with Tn libraries. Please edit for clarity.

Lines 193-197 (discussion for supplemental fig 2) - I was initially confused since the figure shows 3 batches, but the text here implies only 2. Then I remembered there was an initial test batch. To make it clearer to the reader, start the paragraph with something along the lines of "After settling on a data screening and analysis strategy using the first 11 plates, we then processed a second batch...."

Supplemental data 2 - This table indicates mutant number and then well number, but there is no plate ID indicated. From a strain and database management standpoint, Plate ID would be very important here.

Line 205 - Figure 2 is mentioned, but no discussion of the figure is provided.

Figure 2E - I'm not sure I understand the point of this panel. What additional information does this provide? Also, the color scheme for the genes and IG regions are not consistent with that of Panel 2A

Line 218 - Did you examine the locations of these insertions within the genes initially thought to be essential? Is it possible they were towards the end of the gene, leaving it functional or partially functional, indicating it could still be essential?

The methods is lacking a section describing the data analysis pipeline used for analyzing the sequencing results.

Reviewer #2 (Public repository details (Required)):

Sequencing reads should be released prior to publication.

Reviewer #2 (Comments for the Author):

In this present study, the authors have indexed a previously generated Transposon mutagenesis library in Group B Streptococcus strain A909. After screening and indexing, the authors have a library of 878 insertional mutants in known ORFs and an additional 253 insertional mutants in intergenic regions. The authors are currently making this entire library publicly available so that other researchers can utilize this important resource for GBS related research.

This work is of clear importance and will undoubtedly serve as a critical resource for future research related to GBS due primarily to its availability to the larger research community. I applaud the authors for their transparency and commitment to resource availability.

The primary concerns regarding the manuscript revolve around the saturation of the library. The GBS genome is predicted to have ~2000 genes, with this library covering approximately 50% of the entire genome. Given the likely saturation of the original

pooled library, this indicates that there was insufficient colony picking when creating the indexed library. While the authors picked over 8000 colonies, to reach saturation you would likely need to pick over double this number of colonies. (For example, the Nebraska Transposon Mutant Library required picking over 20,000 mutants to reach saturation of the genome MRSA genome, which is approximately the same size as GBS) Although the authors note that the identification of new mutants are reaching a plateau at this level, this is to be expected when considering a Poisson distribution. Although the authors note the financial limitations with performing this, I would strongly suggest additional mutant picking to try to saturate more of the genome.

While the authors highlight that this pooled library has been utilized in previous studies to make novel findings, the present study only validates the utility of the indexed library with known virulence pathways. It would highlight the novelty of this indexed library if it were screened to identify novel genes for known pathogenesis characteristics. For example, screening for novel genes associated with biofilm formation, attachment, intracellular survival etc.

Lastly, I would be remiss if I failed to mention that A909 is not necessarily representative of all GBS isolates. Due to its highly invasive nature and continuous culture in the laboratory, it is less likely to be representative of most clinical isolates. That being said its hyperinvasive nature will allow it to be useful for identifying key genes associated with invasive disease. It may be valuable to discuss the unique nature of this strain and justify why this strain was picked instead of a more relevant isolate.

Staff Comments:

Preparing Revision Guidelines

Please return the manuscript within 60 days; if you cannot complete the modification within this time period, please contact me. If you do not wish to modify the manuscript and prefer to submit it to another journal, please notify me of your decision immediately so that the manuscript may be formally withdrawn from consideration by Microbiology Spectrum.

In this present study, the authors have indexed a previously generated Transposon mutagenesis library in Group B Streptococcus strain A909. After screening and indexing, the authors have a library of 878 insertional mutants in known ORFs and an additional 253 insertional mutants in intergenic regions. The authors are currently making this entire library publicly available so that other researchers can utilize this important resource for GBS related research.

This work is of clear importance and will undoubtedly serve as a critical resource for future research related to GBS due primarily to its availability to the larger research community. I applaud the authors for their transparency and commitment to resource availability.

The primary concerns regarding the manuscript revolve around the saturation of the library. The GBS genome is predicted to have ~2000 genes, with this library covering approximately 50% of the entire genome. Given the likely saturation of the original pooled library, this indicates that there was insufficient colony picking when creating the indexed library. While the authors picked over 8000 colonies, to reach saturation you would likely need to pick over double this number of colonies. (For example, the Nebraska Transposon Mutant Library required picking over 20,000 mutants to reach saturation of the genome MRSA genome, which is approximately the same size as GBS) Although the authors note that the identification of new mutants are reaching a plateau at this level, this is to be expected when considering a Poisson distribution. Although the authors note the financial limitations with performing this, I would strongly suggest additional mutant picking to try to saturate more of the genome.

While the authors highlight that this pooled library has been utilized in previous studies to make novel findings, the present study only validates the utility of the indexed library with known virulence pathways. It would highlight the novelty of this indexed library if it were screened to identify novel genes for known pathogenesis characteristics. For example, screening for novel genes associated with biofilm formation, attachment, intracellular survival etc.

Lastly, I would be remiss if I failed to mention that A909 is not necessarily representative of all GBS isolates. Due to its highly invasive nature and continuous culture in the laboratory, it is less likely to be representative of most clinical isolates. That being said its hyperinvasive nature will allow it to be useful for identifying key genes associated with invasive disease. It may be valuable to discuss the unique nature of this strain and justify why this strain was picked instead of a more relevant isolate.

Reviewer #1 (Comments for the Author):

General Comments

The authors used a previously developed transposon mutant library that had been used in prior TN-seq experiments to generate an indexed transposon mutant library for use in future genetic studies in GBS. This paper presents an extremely useful tool to be used by the research team and is open to the broader scientific community. As a proof of concept, the authors validated phenotypes of a select set of mutants with Tn insertions in previously studied genes. Although this was meant purely as a proof of concept, it would have been more impactful if mutants in genes that had not been thoroughly studied previously had also been examined to further show the usefulness of this tool and assign functions to genes. Additionally, certain portions of the manuscript need further clarification or additional details. More specifically:

Lines 42-43: in the sentence "while the library contains....." What exactly are you referring to here? Are you saying very few have growth defects? If so, this would be expected. Please clarify.

Yes, we do mean essential or near-essential genes, which are expected to have growth defects. We changed the terminology to be (hopefully!) clearer. While the sparsity of insertions in genes with a high-contribution to fitness (essential or near-essential) is expected, we think it is worth stating this at the outset.

In the introduction of bacterial mutant libraries (lines 74-78), or in another section of the introduction, please add information regarding transposon mutant libraries in GBS and the lack of an indexed library and its usefulness to fully explain to the reader the need for this paper.

Thank you. We added a sentence stating the current lack of publicly available GBS indexed mutant libraries, and the negative effect of this absence on the pace of GBS research.

Please provide more information on the background genetics of A909 such as: Why did you choose this particular strain? How large is the genome? How many genes, etc to determine which percentage of the genome and number of genes were hit with TNs in this library.

A909 has a 2,127,839-bp chromosome, first sequenced by TIGR in 2005. The current NCBI annotation posits 2,173 genes. We have added details of the strain's metrics, history, and the motivation for its choice as the background for this library in the Discussion section.

Line 160 mentions previously recognized variation from MmeI digestion, but no reference is provided.

We added a reference for Callahan et al. "Structure of type IIL restriction-modification enzyme MmeI in complex with DNA has implications for engineering new specificities," PLOS Biology (2016). The paper discusses how MmeI-family enzymes do exhibit variability in their cut site. This matches our experience, where—following adapter ligation, PCR, and sequencing—we find evidence of MmeI cutting one base away from its predicted 20-nt range.

Several areas of the paper are divided into rather short paragraphs, making the flow choppy and occasionally causing confusion. For example, in the paragraph for lines 173-176, you mention two unexpected findings, but then the paragraph ends leading the reader to first think you aren't going to mention them and then guess that what is discussed in the next paragraph are the unexpected findings.

We merged those two short paragraphs, which we agree were choppy when divided. We have reviewed and revised throughout to try to make the article read smoothly.

Line 194 - I understand what you are trying to say by "slowing rate of new insertion discovery" but this phrasing is odd and could be confusing for someone less familiar with Tn libraries. Please edit for clarity.

We have rewritten this sentence to improve clarity. Thank you.

Lines 193-197 (discussion for supplemental fig 2) - I was initially confused since the figure shows 3 batches, but the text here implies only 2. Then I remembered there was an initial test batch. To make it clearer to the reader, start the paragraph with something along the lines of "After settling on a data screening and analysis strategy using the first 11 plates, we then processed a second batch...."

We clarified this section as suggested. Thank you!

Supplemental data 2 - This table indicates mutant number and then well number, but there is no plate ID indicated. From a strain and database management standpoint, Plate ID would be very important here.

The revision of Supplemental Data Table 2 includes plate ID labels and accessions for acquisition from BEI.

Line 205 - Figure 2 is mentioned, but no discussion of the figure is provided.

We have added additional details about the panels shown in Figure 2.

Figure 2E - I'm not sure I understand the point of this panel. What additional information does this provide? Also, the color scheme for the genes and IG regions are not consistent with that of Panel 2A

Fair point. We removed Figure 2E. Thanks.

Line 218 - Did you examine the locations of these insertions within the genes initially thought to be essential? Is it possible they were towards the end of the gene, leaving it functional or partially functional, indicating it could still be essential?

We had not examined this, reviewer, and it turns out that you are exactly right. Thank you for the suggestion! These 11 insertions are, in fact, all at the very end of these essential protein coding sequences (range 93-99+%, median 99%). So, that answers that. We modified the text to provide this explanation.

The methods is lacking a section describing the data analysis pipeline used for analyzing the sequencing results.

This section has been added/expanded in the revised draft.

Reviewer #2 (Public repository details (Required)):

Sequencing reads should be released prior to publication.

The sequencing reads are uploaded to SRA and we will make them publicly available upon acceptance for publication.

Reviewer #2 (Comments for the Author):

In this present study, the authors have indexed a previously generated Transposon mutagenesis library in Group B Streptococcus strain A909. After screening and indexing, the authors have a library of 878 insertional mutants in known ORFs and an additional 253 insertional mutants in intergenic regions. The authors are currently making this entire library publicly available so that other researchers can utilize this important resource for GBS related research.

This work is of clear importance and will undoubtedly serve as a critical resource for future research related to GBS due primarily to its availability to the larger research community. I applaud the authors for their transparency and commitment to resource availability.

The primary concerns regarding the manuscript revolve around the saturation of the library. The GBS genome is predicted to have ~2000 genes, with this library covering approximately 50% of the entire genome. Given the likely saturation of the original pooled library, this indicates that there was insufficient colony picking when creating the indexed library. While the authors picked over 8000 colonies, to reach saturation you would likely need to pick over double this number of colonies. (For example, the Nebraska Transposon Mutant Library required picking over 20,000 mutants to reach saturation of the genome MRSA genome, which is approximately the same size as GBS) Although the authors note that the identification of new mutants are reaching a plateau at this level, this is to be expected when considering a Poisson distribution. Although the authors note the financial limitations with performing this, I would strongly suggest additional mutant picking to try to saturate more of the genome.

Thank you for your thoughtful response to our work. We, too, would have liked to achieve a larger collection of unique GBS mutations. The plot of new gene mutations per-well-sequenced can be modeled with a lognormal curve, as shown below. This curve (with dashed 95% confidence intervals indicated) has an asymptote of ~1273 mutants, or around 400 additional gene mutants more than our library. However, reaching this degree of saturation would require processing around 50,000 colonies, or approximately five times more than the number we already processed. We remain a small laboratory with limited resources, and at this time we cannot devote the person-hours and materials necessary for this kind of undertaking. We would not exclude the possibility of a later library expansion, but since it's not possible for us right now we decided to release the work we have done, believing that it would be useful even if the fullest potential of our approach was not currently realizable.

While the authors highlight that this pooled library has been utilized in previous studies to make novel findings, the present study only validates the utility of the indexed library with known virulence pathways. It would highlight the novelty of this indexed library if it were screened to identify novel genes for known pathogenesis characteristics. For example, screening for novel genes associated with biofilm formation, attachment, intracellular survival etc.

Thank you for the suggestion. We have added additional results of a screen for biofilm changes among mutants with transposon insertions in genes that encode surface-trafficked proteins (see Figure 5, which is new).

Lastly, I would be remiss if I failed to mention that A909 is not necessarily representative of all GBS isolates. Due to its highly invasive nature and continuous culture in the laboratory, it is less likely to be representative of most clinical isolates. That being said its hyperinvasive nature will allow it to be useful for identifying key genes associated with invasive disease. It may be valuable to discuss the unique nature of this strain and justify why this strain was picked instead of a more relevant isolate.

Yes, we understand this comment. Please see our response to reviewer 1, above, who also commented on A909. We have added additional rationale to the revised manuscript, recognizing—of course—that there is no single “perfect” strain with which to create such a library.

September 29, 2023

Dr. Thomas Alexander Hooven
University of Pittsburgh
Pediatrics
4401 Penn Ave.
Rangos Bldg #8128
Pittsburgh, PA 15217

Re: Spectrum02046-23R1 (A group B Streptococcus indexed transposon mutant library to accelerate genetic research on an important perinatal pathogen)

Dear Dr. Hooven:

I am very pleased to inform you that your manuscript has been accepted for publication in Microbiology Spectrum. The reviewer's have indicated that you have addressed all of their concerns and the manuscript is much improved. Consequently, I am forwarding it to the ASM Journals Department and you will be notified when your proofs are ready to be viewed.

Sincerely,

Shannon Manning
Editor, Microbiology Spectrum
